# Bioavailability Assessment of Yarrow Phenolic Compounds Using an In Vitro Digestion/Caco-2 Cell Model: Anti-Inflammatory Activity of Basolateral Fraction

**DOI:** 10.3390/molecules27238254

**Published:** 2022-11-26

**Authors:** Marisol Villalva, Laura Jaime, María de las Nieves Siles-Sánchez, Susana Santoyo

**Affiliations:** Institute of Food Science Research (CIAL), Universidad Autónoma de Madrid (CEI UAM+CSIC), 28049 Madrid, Spain

**Keywords:** *Achillea millefolium*, bioaccessibility, Caco-2 absorption, in vitro digestion, phenolic compounds

## Abstract

In this study, a combined in vitro digestion/Caco-2 model was performed with the aim to determine the phenolic compounds bioavailability of two yarrow extracts. HPLC-PAD characterisation indicated that the main components in both extracts were 3,5-dicaffeoylquinic acid (DCQA) and luteolin-7-*O*-glucoside. Analyses after the simulated digestion process revealed that phenolic composition was not affected during the oral phase, whereas gastric and intestinal phases represented critical steps for some individual phenolics, especially intestinal step. The transition from gastric medium to intestinal environment caused an important degradation of 3,5-DCQA (63–67% loss), whereas 3,4-DCQA and 4,5-DCQA increased significantly, suggesting an isomeric transformation within these caffeic acid derivatives. However, an approx. 90% of luteolin-7-*O*-glucoside was recovered after intestinal step. At the end of Caco-2 absorption experiments, casticin, diosmetin and centaureidin represented the most abundant compounds in the basolateral fraction. Moreover, this fraction presented anti-inflammatory activity since was able to inhibit the secretion of IL-1*β* and IL-6 pro-inflammatory cytokines. Thus, the presence in the basolateral fraction of flavonoid-aglycones from yarrow, could be related with the observed anti-inflammatory activity from yarrow extract.

## 1. Introduction

*Achillea millefolium* L. (yarrow) is a flowering plant traditionally used in the treatment of digestive and hepatobiliary disorders, inflammation, and diabetes [1]. Recent reports indicated that *Achillea* genus presented important biological activities, such as antioxidant, anti-inflammatory and antitumor activities [2]. Most health benefits of aqueous and alcoholic yarrow extracts have been associated with its composition in phenolic compounds, mainly phenolic acids (caffeic acid derivatives), and flavonoids (luteolin, apigenin and quercetin derivatives) [1,3]. Thus, Trumbeckaite et al. [4] related the antioxidant properties of an *Achillea millefolium* hydroalcoholic extract with the presence of luteolin and chlorogenic acid in the extract, and in a lesser extent with rutin and luteolin-7-*O*-glucoside. Pereira et al. [5] also reported that an *A. millefolium* hydroethanolic extract, containing 3,5-*O*-dicaffeoylquinic acid, 5-*O*-caffeoylquinic acid, luteolin-*O*-acetylhexoside and apigenin-*O*-acetylhexoside as main phenolic compounds, inhibited the growth of human tumour cell lines. Furthermore, both essential oils and hydroethanolic yarrow extracts have demonstrated anti-inflammatory properties, causing the inhibition of nitric oxide (NO) production and IL-8 secretion in vitro [6,7].

Nevertheless, after oral consumption, phenolic compounds must be bioavailable in order to perform their potential health benefits. The bioavailability is dependent upon the stability of the compound during gastrointestinal digestion, its release from the food-matrix and the efficiency of its intestinal absorption. In yarrow, the assessment of mineral and vitamins bioaccessibility has been performed [8], but studies for phenolic compounds are still scarce. The stability of phenolic compounds during the gastrointestinal digestion is strongly influenced by their chemical structure, since phenolics present a different sensitivity to pH variations and digestive enzymes activity [9,10]. Moreover, the stability of phenolic compounds under gastrointestinal conditions highly depends on the nature of the matrix in which these compounds are included [11]. Thereby, Lingua et al. [12] reported that phenolic acids and quercetin were the most resistant polyphenols in white grape after a simulated digestion. However, Ortega-Vidal et al. [13] indicated that caffeoylquinic acids in herbal infusions were highly reduced by gastrointestinal digestion (approx. 10% remain). Moreover, Spínola et al. [14] carried out extracts of *Rumex maderensis* and reported that the degradation of different phenolic classes after digestion varied within morphological parts employed (leaves, flowers, and stems). Thus, flavanols were the most stable compounds, although in flowers presented a reduction of 29.7% against 40.4% in stems. Hydroxycinnamic acids from leaves and flowers, presented a similar degradation rate (approx. 56.5%), meanwhile in stems extracts hydroxycinnamic acids were very unstable (71.8% reduction).

After gastrointestinal digestion, the intestinal absorption of phenolic compounds has also been reported to be highly influenced by the phenolic compounds chemical structure. Bowles et al. [15] studied the intestinal transport across Caco-2 cell monolayer of nine phenolic acids found in an aqueous extract of *Athrixia phylicoides*, concluding that *p*-coumaric acid presented the highest transport. Besides, Wu et al. [16] reported that the absorption of caffeic acid was higher than chlorogenic acid in the Caco-2 model as well as in rat jejunum. Therefore, the use of an in vitro digestion/Caco-2 cell culture model has been proposed by several authors as an economical and useful alternative to the in vivo analysis, in order to investigate the bioavailability of phenolic compounds [12,17].

Concerning phenolic compounds extraction, several studies proposed the ultrasound-assisted extraction (UAE) as an adequate technique to obtain phenolic compounds from vegetal matrices [18,19]. In this regard, UAE has been reported to reduce extraction time and solvent consumption, as well as to maximizing the recoveries of bioactive compounds [20]. However, sometimes it is difficult to obtain highly concentrated extracts using only UAE, due to complexity of vegetable raw materials. Therefore, the use of adsorption resins (e.g., XAD-2, XAD-7, XAD-16 and Oasis HLB) has been successfully employed as a tool for selective enrichment of phenolic compounds from plant material [21,22]. The aim of this work was to study the bioavailability of yarrow phenolic compounds, by using a combined in vitro digestion/Caco-2 cell model. In addition, the influence of phenolics compounds concentration in the matrix on their bioavailability was also determined. Besides, the biological activity of Caco-2 basolateral fraction, in terms of anti-inflammatory activity was measured.

## 2. Results and Discussions

### 2.1. Influence of In Vitro Gastrointestinal Steps on Phenolic Composition and Antioxidant Activity of the Extracts

Phenolic compounds identification of yarrow extract (YE) and yarrow phenolic enriched-extract (EE) was performed by HPLC-PAD-ESI-QTOF-MS allowing the identification of 49 phenolic compounds (Appendix A). These results were in accordance with similar reported YE composition [23]. The quantitative analysis in both extracts (YE and EE) before and after the three-steps digestion process (oral, gastric, and intestinal) were shown in Table 1 and Table 2, respectively. As can be observed, both extracts presented a similar behaviour during the gastrointestinal process. In general, the phenolic composition of both yarrow extracts was not affected during the oral phase, whereas gastric and intestinal phases, especially intestinal one, resulted as critical steps for some individual phenolic compounds. Appendix A shows the base peak chromatogram of the EE before and after intestinal digestion, where the major differences can be observed.

Chlorogenic acid (CGA), the most abundant mono-caffeoylquinic acid in both extracts, was stable under oral and gastric conditions, but showed a loss of about 17% at the end of the intestinal step. However, it should be highlighted the higher quantity of neochlorogenic and cryptochlorogenic acids measured after intestinal step, whose increase could be attributed to CGA isomerization. This behaviour was also found by Bouayed et al. [24], who reported that CGA was stable to gastric conditions but degraded (between 23–41%) during intestinal digestion, with partial isomerisation to neochlorogenic and cryptochlorogenic acids. Yu et al. [25] also reported an important bioaccessibility (68.39–91.34%) after digestion process for chlorogenic acid obtained from mulberry leaves.

With respect to dicaffeoylquinic acids (DCQAs), these compounds seem to be stable under oral conditions in both extracts. Gastric conditions mainly affected 3,5- and 4,5-DCQAs with a significant loss of approx. 20%. The transition from gastric medium to intestinal environment caused an important degradation of 3,5-DCQA (63–67% loss), whereas 3,4-DCQA and 4,5-DCQA significantly increased their quantity after intestinal step. This increment could be related with isomerization processes among different DCQAs at intestinal pH. Moreover, at the end of the intestinal step, the sum of all DCQAs represented the 90% of these compounds in the undigested extract. D’Antuono et al. [26] previously described that 3,5-DCQA (pure individual compound) gastrointestinal digestion produced a higher isomerization effect with the presence of 3,4-DCQA and 4,5-DCQA.

Both extracts, YE and EE, also presented an important quantity of flavonoids, either in glycosylated or in aglycone form. Among the glycosylated forms, luteolin-7-*O*-glucoside, the most abundant compound within flavonoids group, was stable to oral digestion but gastric conditions produced a decrease of approx. 20%. However, this compound increased up to 87–90% at the end of intestinal step. Gutiérrez-Grijalva et al. [27] also found that in digestion process, the quantity of luteolin-7-*O*-glucoside decreased after gastric step but increased at the end of intestinal step. They indicated that the loss of luteolin-7-*O*-glucoside after gastric phase could be related, in addition to pH changes, to a possible interaction between the compound and gastric enzymes that render it undetectable in chromatographic analysis. Regarding aglycones, luteolin was also stable to oral step but hardly affected by gastric conditions (approx. 50% loss). Moreover, luteolin registered an increased when intestinal phase ended. This behaviour was observed in other aglycones such as casticin and centaureidin. According with previous results, this effect could be also related with possible interactions between digestive enzymes and phenolics, as was detected for luteolin-7-*O*-glucoside.

Digestion effect on total phenolic content (TPC) and antioxidant activity for both extracts is shown in Table 3. During digestion process, the amount of TPC only decreased slightly for both extracts (between 7–13%). Regarding antioxidant activity, this was not significantly affected during oral phase, however, stomach and intestinal phases resulted in critical steps for this activity (26–40% decrease). This loss of antioxidant activity could be related with the losses registered in some phenolic compounds, such as luteolin and its glucosylated derivatives, since these compounds have been reported to present an important antioxidant activity [28]. However, the isomerization effect occurred in some compounds (i.e., CGA and DCQAs) could also be related. Shang et al. [29] indicated that among DCQA isomers from a *L. fischeri* leaves ethanolic extract, 3,5-DCQA presented the highest radical scavenging activity.

### 2.2. Caco-2 Cells Transport Experiments

In order to investigate the potential bioavailability of digested yarrow phenolic compounds, their intestinal uptake was evaluated using Caco-2 cells monolayers at 2, 4 and 6 h. Due to EE digested extract presented a 2–3 fold superior concentration of phenolic compounds, this extract was selected to carry out the transport experiments. The cytotoxicity assays, performed by the MTT method, indicated that 40 µL/mL of digested EE was the maximum concentration that did not affect the cell viability during 6 h (data not shown). Thus, the concentration of EE phenolic compounds detected in the apical compartment, cellular monolayer, and basolateral compartment after 2, 4 and 6 h of transport experiments are shown in Table 4. After 2 h of incubation, 11 phenolic compounds were identified in the cell monolayer, mainly flavonoid aglycones (casticin, diosmetin and centaureidin) and DCQAs isomers (3,4-DCQA and 3,5-DCQA). The concentration of those compounds in cell monolayer decreased after 4 and 6 h of experiment. Regarding the basolateral compartment, after 2 h of incubation, casticin was the main compound, followed by 3,4 and 3,5-DCQAs. Data obtained after 4 h showed an increase in casticin, diosmetin and centaureidin concentration in the basolateral compartment, meanwhile the amount of 3,5-DCQA remains constant and 3,4-DCQA slightly decreased. Successively, an increment in the quantity of casticin, diosmetin and centaureidin continued until 6 h of experiment, while neither 3,4 nor 3,5-DCQAs were detected at that time.

Casticin (a methoxylated flavonol) was the most abundant compound in the basolateral fraction (after 6 h, a 41.7% from digested extract). The apparent permeability coefficients (*P_app_*) for casticin presented a maximum value at 2 h (*P_app_* = 16.7 ± 0.1 × 10^−6^ cm s^−1^) in comparison with 4 h and 6 h (*P_app_* = 10.9 ± 0.1 × 10^−6^ and 10.2 ± 0.3 × 10^−6^ cm s^−1^, respectively). These results suggested that casticin permeability was time-dependent and transported across the Caco-2 monolayers with a faster rate at a shorter incubation time. In spite of in vitro studies of casticin’s permeability are still scare; Piazzini et al. [30] reported a casticin’s *P_app_* value of 8.1 ± 0.9 × 10^−6^ cm s^−1^ across Caco-2 cells, after 4 h incubation with a *Vitex agnus-castus* extract.

Diosmetin’s uptake also increased with incubation time. Surprisingly, the sum of diosmetin amount in cell monolayer and basolateral fraction (at 2, 4 or 6 h), was higher than the concentration of this compound initially placed in the apical side (0.84 ± 0.1 mg/L of digested extract). Thus, considering that diosmetin is the 4′-methyl derivative of luteolin, the detected increment could be originated from the metabolism of luteolin (aglycone) and/or luteolin glycosylated-derivatives presented in the digested extract. The occurrence of diosmetin as a principal methylated metabolite from luteolin was reported in rats [31].

Centaureidin represented the third most abundant compound in the basolateral fraction after 6 h. This methylated flavonol, also showed a time-dependent absorption through Caco-2 cell monolayer, being more rapidly transported at earlier incubation time (*P_app_* at 2 h = 7.0 ± 0.4 × 10^−6^ cm s^−1^). To the best of our knowledge, no previous studies had been reported for centaureidin’s in vitro absorption. In general, a passive transcellular diffusion through Caco-2 monolayer could be related with casticin, diosmetin and centaureidin absorption. Nevertheless, interactions of diosmetin with selected transporters such as multidrug resistance-associated protein isoform 1 (MRP1) and monocarboxylate transporter isoform 1 (MCT1), also expressed in Caco-2 cells, have been previously described [32].

The amounts of 3,4-DCQA and 3,5-DCQA detected in the basolateral compartment after 2 and 4 h of incubation were quite smaller with respect to their amounts in the apical side, showing a *P_app_* calculated values (at 4 h) of 1.0 ± 0.1 × 10^−6^ cm s^−1^ for 3,4-DCQA and 2.2 ± 0.1 × 10^−6^ cm s^−1^ for 3,5-DCQA. Similarly, Zhou et al. [33] indicated that DCQAs showed, after 4 h, *P_app_* values of approx. 2.5 × 10^−6^ cm s^−1^. However, at 6 h, unexpectedly no DCQAs isomers were detected in basolateral fraction (Table 4). In consistence with this result, D’Antuono et al. [26], did not also detect any DCQAs isomer in the basolateral side, but coumaric and caffeic acids were found in this fraction, suggesting a cellular metabolism activity. However, in our results neither caffeic nor coumaric acids were found in the basolateral fraction at detectable amounts with the analytical technique employed. Regarding Caco-2 transport, Zhou et al. [33] described DCQAs absorption mainly by passive diffusion via paracellular pathways, although some interactions of DCQAs with certain transporters were also reported in vitro [34]. In that context, when evaluating the absorption of a complex plant-extract, we would have to consider that certain phenolics may act like substrates or inhibitors of some transporters expressed in Caco-2 cells, thus, they could act as permeability modifiers for other compounds [35].

### 2.3. Anti-Inflammatory Activity of Caco-2 Cells Basolateral Fraction

Basolateral fraction recovered at 6 h was used to carry out the anti-inflammatory assays, using THP-1 macrophages (stimulated via LPS) to quantify the pro-inflammatory cytokines secretion in the medium. In addition, the basolateral fraction from control digestion (digestion fluids without extract) was also tested. These results are shown in Figure 1. As can be observed, after 24 h the stimulated macrophages (positive control) revealed a significant release of the three pro-inflammatory cytokines, TNF-*α*, IL-1*β* and IL-6, compared to non-stimulated cells (negative control). Previous experiments to assess the cytotoxicity of the basolateral fraction indicated that 20 µL/mL did not compromise the macrophages viability (data not shown). Thus, when THP-1 macrophages were incubated with LPS in presence of 10 and 20 µL/mL of the basolateral media, TNF-*α* secretion was not modified, compared with the levels obtained in absence of the extracts (Figure 1A). In contrast, a significant reduction of IL-1*β* secreted was observed in presence of both concentrations of basolateral fraction, approx. 30% and 40% for 10 and 20 µL/mL (Figure 1B). The IL-6 release was also supressed approx. 25% when applied 20 µL/mL of basolateral fraction (Figure 1C).

Thus, the basolateral fraction from EE exhibited a moderate inhibition of IL-1*β* and IL-6 cytokines. Considering that this fraction was mainly composed by casticin, diosmetin and centaureidin, these flavonoids could be related, at least partially, with the anti-inflammatory activity. Casticin was shown to decrease the production of pro-inflammatory cytokines, such as IL-1*β*, IL-6, and TNF-*α* in RAW 264.7 cells treated with LPS [36]. Moreover, diosmetin also reduced the generation of pro-inflammatory mediators like NO, TNF-α in adipocytes and macrophages, and IL-1*β* e IL-6 in rheumatoid arthritis fibroblast [37]. Finally, centaureidin has been also effectively inhibited expression of COX-1 and COX-2 enzymes related with the inflammatory response [38]. Nevertheless, the influence of other compounds, including those found in minor concentrations or even those non-detected metabolites of phenolic compounds, cannot be ruled out.

## 3. Materials and Methods

### 3.1. Yarrow Extract and Yarrow Phenolic Compounds-Enriched Extract Obtention

Upper-dried inflorescences of yarrow were obtained from a local supplier (Plantafarm S.A., León, Spain). The sample was ground (Premill 250, Leal S.A., Granollers, Spain) and sieved to diminish its particle size (<500 μm). YE was obtained by ultrasound assisted extraction accordingly to Villalva et al. [23]. Briefly, the ground yarrow was soaked with pure ethanol (plant/solvent 1:10, *w*/*v*) and conducted to extraction (30 min, ≤40 °C) in a Branson 450 ultrasonic device (Branson Ultrasonics, Danbury, CT, USA). The solvent was removed in a vacuum rotary evaporator (35 °C) (IKA RV 10, VWR, Madrid, Spain) to obtain a dry extract.

In order to obtain the EE, a fractionation process was conducted using XAD-7HP macroporous resins (Sigma-Aldrich, St. Louis, MO, USA) packed in a glass column (bed volume, BV, 225 mL) and a mixture of ethanol:water (80:20, *v*/*v*) as elution solvent. A volume (70 mL) of YE (15 mg/mL final concentration) was placed in the column. After 1 h to allowing the absorption equilibrium, a water-washing step was required (2 BV) to later recover the phenolic compounds using 80% ethanol (3 BV at 2 BV/h). At the end, the ethanol was evaporated in a rotary evaporator and freeze-dried to remove the water. The extracts were kept at −20 °C until analysis. All experiments were done by triplicate.

A graphical flowchart summarizing the main steps of the experimental procedure applied is provided in Figure 2.

### 3.2. HPLC-PAD Phenolic Compounds Analysis

Phenolic compounds analysis was performed using an HPLC 1260 Infinity series system with a photodiode-array detector (PAD) (Agilent Technologies Inc., Santa Clara, CA, USA). Both YE and EE dried extracts were dissolved in ethanol or ethanol:water (50:50, *v*/*v*), filtered (PVDF, 0.45 μm) and analysed by HPLC-PAD. Chromatographic separation was carried out with a reverse phase ACE Excel SuperC18 column (ACT, Aberdeen, Scotland), equipped with a guard-column of the same material, according to the methodologic conditions described by Villalva et al. (2018) [22]. Identification of phenolic compounds was based on HPLC-PAD-ESI-QTOF-MS/MS analysis by following the Villalva et al. (2021) [23] procedure. Quantification was performed by HPLC-DAD according to the calibration curve established of each authentic phenolic standard (HPLC purity ≥ 95%). Calibration curves were also used for the quantification of phenolic compounds with unavailable commercial standard, following their chemical similarity, e.g., apigenin 7-*O*-glucoside was used for apigenin derivative, and chlorogenic acid curve for caffeoylquinic acid isomers.

### 3.3. Determination of Total Phenolic Content (TPC) and Antioxidant Activity

Total phenolic content was determined by Folin-Ciocalteau reagent as described by Singleton et al. [39]. The results were expressed as mg of gallic acid equivalents (GAE)/g extract. DPPH (2,2-diphenyl-1-picrylhydrazyl, Sigma-Aldrich, Madrid, Spain) free radical methodology was used to evaluate the antioxidant activity according to Brand-Williams et al. [40]. The results were expressed as TEAC value (mmol Trolox/g of extract or mmol Trolox/L of digested extract). All analyses were done in triplicate.

### 3.4. In Vitro Gastrointestinal Digestion

YE and EE were subjected to a three steps digestion process [22]. Briefly, 5 mL of extract solution (20 mg/mL) with 0.1 mL α-amylase from human saliva (9.3 mg in Cl_2_Ca 1 mM) (Type XIII-A, Sigma-Aldrich, St. Louis, MO, USA) were stirred for 2 min in a titrator Titrino Plus 877 at 37 °C (Methrom AG, Herisau, Switzerland) (oral phase). Then, 25 mL of a gastric solution (pH 2.0 ± 0.5) containing 127 mg of pepsin from porcine gastric mucosa (536 U/mg, Sigma-Aldrich, St. Louis, MO, USA) was added and incubated for 1 h (gastric phase). After gastric digestion, pH was adjusted to 7.5 ± 0.5 by addition of 10 mL intestinal solution composed by 5.3% (*v*/*v*) of NaOH 0.1 M, 1.4% (*v*/*v*) of NaCl 3.25 M, 0.5% of CaCl_2_ 325 mM and 2.8% (*v*/*v*) of a pancreatic-bile extract solution (9.3 mg pancreatin (4 × USP) and 115.7 mg bile salts in 10 mM trizma-maleate buffer), allowing stirring for 2 h to simulate intestinal phase. When digestion finished, the solutions were immediately cooled and filtered (0.45 μm, PVDF) to conduct the HPLC-PAD analysis, TPC and antioxidant activity assays. Additionally, digestion steps, without yarrow sample addition, were also carried out as control digestion.

### 3.5. Caco-2 Cells Culture and Transport Experiments

Maintenance conditions for Caco-2 cell line (ATCC, Manassas, VA, USA), as well as the cell viability experiments, were followed as previously described by Villalva et al. [22]. To assess the transport assays, Caco-2 cells (density 3 × 10^5^ cells/insert) were seeded in polyester Transwell^®^ inserts (24 mm diameter, 0.4 μm pore size, Corning Life Science) and cultured for 21 days at 37 °C (5% CO_2_). The day of the transport experiments, the inserts were carefully washed with Phosphate Buffer Solution (PBS) (Gibco, Paisley, UK) and filled with 1.5 mL (apical) and 2.6 mL (basolateral) of pre-warmed Dulbecco’s Modified Eagle’s Medium (DMEM) (Lonza, Basel, Switzerland) without phenol red, and a specific volume of digested yarrow extract was added in the apical compartment (extract final dilution 1:25, *v*/*v*). At the end of 2, 4 and 6 h of incubation apical and basolateral supernatants were collected, freeze-dried and stored (−20 °C) until analysis. Cell monolayer integrity was measured before and after the transport assays using an EVOM2 epithelial volt-ohm meter (World Precision Instruments, Hitchin, UK) and only inserts with transepithelial electric resistance (TEER) values > 700 Ωcm^2^ were used. In addition, lucifer yellow (Sigma-Aldrich, Madrid, Spain) permeation was determined to validate the integrity of cell barrier, according to Uchida et al. [41]. To performed the HPLC-PAD analysis, lyophilized samples from apical and basolateral sides, were conducted to extraction with 60:40 ethanol:water (*v*/*v*) (150 μL and 175 μL, respectively) followed by centrifugation (15,000 RPM, 5 min). The supernatants were filtered (0.45 μm, PVDF filters) before HPLC analysis.

Cell monolayers were washed with cold PBS, followed by 500 μL pure ethanol addition. After incubation (4 °C, 30 min), cells were scraped off the membrane, sonicated (5 min) and centrifuged (4500 RPM, 15 min) to recover the supernatant. This process was repeated three times, and finally all supernatants were evaporated until dryness with pure NO_2_. The final residue was re-dissolved in 60:40 ethanol:water (*v*/*v*) (100 μL) and filtered prior HPLC injection.

The apparent permeability coefficient (*P_app_*, cm s^−1^) of each compound detected in the basolateral supernatant was determined according to D’Antuono et al. [26] with the following equation:(1)$$P_{app=}\frac{\left(dC/dt\right)\;V}{C_{o}A}$$
where *dC*/*dt* is the apparent rate of polyphenols transported to the basolateral compartment over the time (μg L^−1^ s^−1^), *V* is the volume of the basolateral compartment (cm^3^); *C_o_* is the initial concentration in the apical compartment (μg L^−1^) and *A* is the surface area of the membrane (cm^2^).

### 3.6. Anti-Inflammatory Assays of Basolateral Fraction from Caco-2 Experiments

Differentiated macrophages from the human monocyte THP-1 cell line (ATCC, Manassas, VA, USA) was used to conduct anti-inflammatory assays according to Villalva et al. [22] with minor modifications. Briefly, THP-1 cells were seeded in 24 well-plate (5 × 10^5^ cells/mL) and differentiated with 100 ng/mL of phorbol 12-myristate 13-acetate (PMA) (Sigma-Aldrich, Madrid, Spain) maintained for 48 h (37 °C, 5% CO_2_). The cytotoxic effect of the basolateral supernatants from Caco-2 over THP-1 macrophages, was determined by the 3-(4,5-dimethylthiazol-2-yl)-2,5-diphenyl tetrazolium bromide assay (MTT) (Sigma-Aldrich, Madrid, Spain). Afterwards, the macrophages were washed and filled with serum-free RPMI medium (Gibco, Paisley, UK) along with a non-toxic concentration of basolateral supernatants and 0.05 μg/mL of bacterial lipopolysaccharide (LPS) (*E. coli* O55:B5, Sigma-Aldrich, Madrid, Spain). After 24 h incubation, the medium was collected and the release of pro-inflammatory cytokines, TNF-*α*, IL-1*β*, and IL-6, was measured by an enzyme-linked immunosorbent assay (ELISA) (BD Biosciences, Aalst, Belgium) according to the manufacturer protocol. Cells with LPS but without basolateral sample, represented the positive control of the immunomodulatory assay; negative control was the non-stimulated cells in absence of basolateral sample. Results were expressed as mean of three determinations ± standard deviation.

### 3.7. Statistical Analysis

Experimental results are expressed as means ± standard deviation (SD). Variance one-way analysis (ANOVA) followed by Fisher’s least significance differences (LSD) test were used to distinguish differences between means at *p* < 0.05. Statgraphics Centurion XVI software (Version 16, Statpoint Technologies Inc., Warrenton, VA, USA) was used for that purpose.

## 4. Conclusions

Phenolic compounds from yarrow showed a great stability at oral step during the simulated digestion, however gastric and intestinal phases caused important modifications. Mostly CGA and DCQAs suffered an isomerization effect after intestinal step. Besides flavonoids, either in their glycosylated or aglycone form, were also reduced after intestinal phase. Casticin, diosmetin and centaureidin were the most abundant compounds found in the basolateral fraction after Caco-2 experiments at 6 h. This fraction also exhibited a certain inhibition of the pro-inflammatory cytokines IL-1*β* and IL-6. Thus, the phenolic composition found in this fraction, mainly methoxylated flavonoids, could be related with the observed bioactivity. Although in vitro results cannot be directly extrapolated to human in vivo conditions, our findings exhibit a potential bioavailability of phenolic compounds present in yarrow extracts.

## Figures and Tables

**Figure 1 molecules-27-08254-f001:**
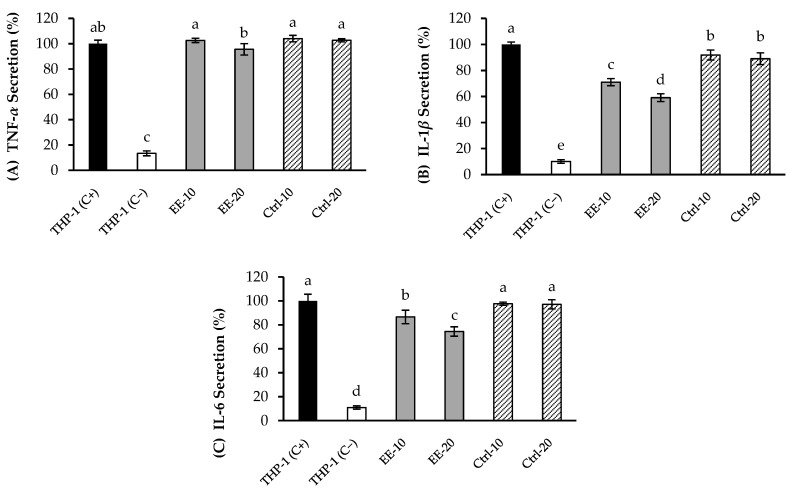
Levels of TNF-*α* (**A**), IL-1*β* (**B**) and IL-6 (**C**) secreted by THP-1 macrophages activated with LPS, in presence of 10 μL/mL (EE-10) and 20 μL/mL (EE-20) of basolateral fraction from yarrow enriched-extract (EE). Positive control (THP-1 C+), cells stimulated with LPS without basolateral sample. Negative control (THP-1 C−), cells stimulated with LPS in contact just with RPMI medium. Control digestion (Ctrl) represents the basolateral supernatant from digested fluids without extract at 10 μL/mL (Ctrl-10) and 20 μL/mL (Ctrl-20). Each bar is the mean of three determinations ± S.D. ^a, b, c, d, e^ Different letters indicate statistical differences among samples (*p* < 0.05) according to Fisher’s least significant difference (LSD) procedure.

**Figure 2 molecules-27-08254-f002:**
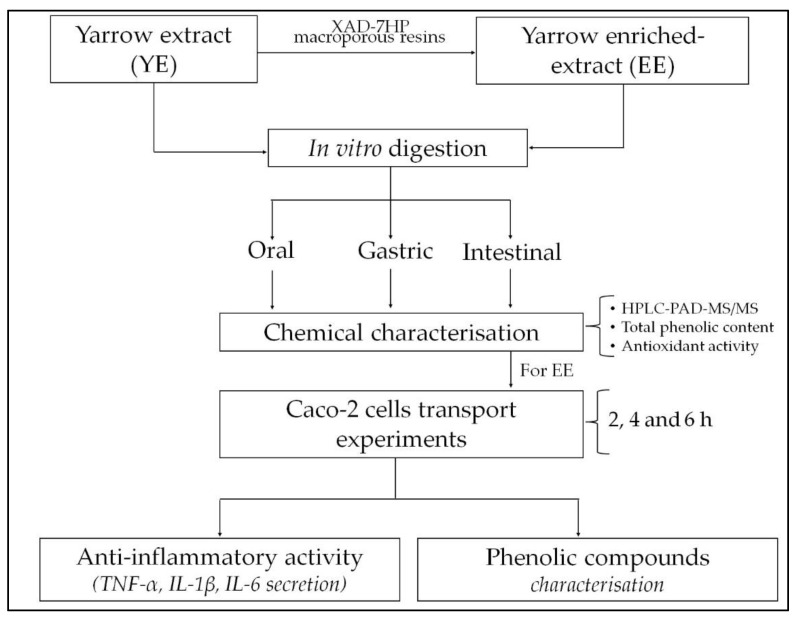
Flowchart summarizing the main experimental procedures for the bioavailability assessment of yarrow phenolic compounds.

**Table 1 molecules-27-08254-t001:** Phenolic composition (mg/g extract) of yarrow extract (YE) before and after oral, gastric and intestinal digestion steps.

Compound	Undigested YE	Oral	Gastric	Intestinal
Neochlorogenic acid	0.24 ± 0.11 ^b^	0.21 ± 0.06 ^b^	0.29 ± 0.09 ^b^	0.56 ± 0.07 ^a^
Protocatechuic acid	0.13 ± 0.10 ^b^	0.12 ± 0.07 ^b^	0.13 ± 0.08 ^b^	0.47 ± 0.12 ^a^
Caftaric acid isomer	0.08 ± 0.03 ^a^	0.08 ± 0.04 ^a^	0.06 ± 0.03 ^ab^	0.04 ± 0.03 ^b^
Caftaric acid	0.07 ± 0.03 ^a^	0.07 ± 0.04 ^a^	0.18 ± 0.09 ^a^	0.13 ± 0.07
Caffeoylquinic acid isomer I	0.39 ± 0.09 ^a^	0.39 ± 0.08 ^a^	0.24 ± 0.08 ^ab^	0.22 ± 0.06 ^b^
Chlorogenic acid	5.67 ± 0.25 ^a^	5.02 ± 0.21 ^ab^	5.90 ± 0.30 ^a^	4.75 ± 0.20 ^b^
Cryptochlorogenic acid	0.13 ± 0.05 ^b^	0.10 ± 0.03 ^b^	0.17 ± 0.04 ^b^	0.75 ± 0.12 ^a^
Vicenin 2	2.11 ± 0.10 ^bc^	2.02 ± 0.10 ^c^	2.45 ± 0.15 ^a^	2.24 ± 0.10 ^ab^
Caffeoylquinic acid isomer II	0.10 ± 0.03 ^a^	0.12 ± 0.04 ^a^	0.10 ± 0.02 ^a^	0.10 ± 0.03 ^a^
Apigenin hexoside-pentoside I	0.46 ± 0.06 ^a^	0.48 ± 0.05 ^a^	0.49 ± 0.06 ^a^	0.43 ± 0.06 ^a^
Caffeic acid	0.34 ± 0.04 ^a^	0.36 ± 0.06 ^a^	0.40 ± 0.05 ^a^	0.42 ± 0.06 ^a^
Schaftoside isomer	1.34 ± 0.10 ^a^	1.32 ± 0.09 ^a^	1.43 ± 0.10 ^a^	1.43 ± 0.12 ^a^
Schaftoside	1.77 ± 0.18 ^ab^	1.61 ± 0.15 ^b^	2.14 ± 0.19 ^a^	2.01 ± 0.16 ^a^
Homoorientin	2.10 ± 0.19 ^a^	1.94 ± 0.12 ^a^	2.20 ± 0.15 ^a^	1.89 ± 0.12 ^a^
Apigenin hexoside-pentoside II	1.04 ± 0.11 ^a^	0.97 ± 0.09 ^a^	1.04 ± 0.10 ^a^	0.98 ± 0.08 ^a^
Luteolin dihexoside I	2.60 ± 0.18 ^ab^	2.32 ± 0.12 ^b^	2.77 ± 0.11 ^a^	2.52 ± 0.11 ^b^
6-hydroxyluteolin-7-*O*-glucoside	2.03 ± 0.12 ^b^	1.97 ± 0.08 ^b^	2.34 ± 0.12 ^a^	1.74 ± 0.09 ^c^
Apigenin dihexoside	0.15 ± 0.09 ^a^	0.16 ± 0.06 ^a^	0.21 ± 0.07 ^a^	0.16 ± 0.04 ^a^
Quercetin hexoside	1.33 ± 0.13 ^a^	1.31 ± 0.08 ^a^	1.10 ± 0.10 ^a^	0.25 ± 0.07 ^b^
Luteolin dihexoside II	0.23 ± 0.04 ^a^	0.23 ± 0.06 ^a^	0.27 ± 0.07 ^a^	0.24 ± 0.04 ^a^
Rutin	1.06 ± 0.07 ^a^	1.08 ± 0.09 ^a^	1.16 ± 0.07 ^a^	1.02 ± 0.09 ^a^
Apigenin hexoside	0.50 ± 0.04 ^a^	0.47 ± 0.07 ^a^	0.57 ± 0.06 ^a^	0.52 ± 0.06 ^a^
Vitexin	0.67 ± 0.07 ^a^	0.61 ± 0.09 ^a^	0.72 ± 0.08 ^a^	0.64 ± 0.07 ^a^
Apigenin hexoside-deoxyhexoside	0.40 ± 0.05 ^a^	0.42 ± 0.04 ^a^	0.25 ± 0.04 ^b^	0.22 ± 0.03 ^b^
Apigenin derivative	2.52 ± 0.12 ^b^	2.49 ± 0.09 ^b^	2.54 ± 0.10 ^b^	2.72 ± 0.11 ^a^
Luteolin-7-*O*-glucoside	8.29 ± 0.28 ^a^	8.12 ± 0.32 ^a^	6.70 ± 0.25 ^c^	7.24 ± 0.33 ^b^
Luteolin-7-*O*-glucuronide	0.72 ± 0.09 ^a^	0.69 ± 0.08 ^ab^	0.57 ± 0.05 ^b^	0.69 ± 0.07 ^ab^
Quercetin hexuronide	0.15 ± 0.03 ^b^	0.12 ± 0.05 ^b^	0.25 ± 0.03 ^a^	0.20 ± 0.04 ^ab^
3,4-Dicaffeoylquinic acid	1.49 ± 0.10 ^b^	1.37 ± 0.08 ^b^	1.42 ± 0.08 ^b^	6.26 ± 0.27 ^a^
Isorhamnetin hexoside I	1.59 ± 0.12 ^a^	1.49 ± 0.09 ^a^	1.00 ± 0.07 ^b^	1.00 ± 0.06 ^b^
1,5-Dicaffeoylquinic acid	1.65 ± 0.11 ^a^	1.66 ± 0.10 ^a^	1.49 ± 0.07 ^ab^	1.37 ± 0.08 ^b^
3,5-Dicaffeoylquinic acid	23.8 ± 1.81 ^a^	22.9 ± 1.13 ^a^	18.8 ± 0.90 ^b^	8.77 ± 0.11 ^c^
Apigenin-7-*O*-glucoside	2.27 ± 0.10 ^a^	2.15 ± 0.07 ^ab^	2.01 ± 0.08 ^b^	1.81 ± 0.09 ^c^
Luteolin-*O*-malonylglucoside	0.53 ± 0.04 ^a^	0.52 ± 0.03 ^a^	0.50 ± 0.04 ^ab^	0.44 ± 0.03 ^b^
4,5-Dicaffeoylquinic acid	4.25 ± 0.20 ^b^	4.05 ± 0.18 ^b^	3.61 ± 0.12 ^c^	11.5 ± 0.51 ^a^
Isorhamnetin hexoside II	0.62 ± 0.06 ^b^	0.60 ± 0.04 ^b^	0.50 ± 0.04 ^c^	1.35 ± 0.10 ^a^
Dicaffeoylquinic acid isomer	0.06 ± 0.01 ^b^	0.05 ± 0.02 ^b^	0.06 ± 0.02 ^b^	0.10 ± 0.01 ^a^
Feruloylcaffeoylquinic acid	0.14 ± 0.03 ^a^	0.12 ± 0.02 ^a^	0.07 ± 0.02 ^b^	0.11 ± 0.03 ^ab^
Tricaffeoylquinic acid	0.36 ± 0.06 ^a^	0.31 ± 0.04 ^ab^	0.09 ± 0.01 ^c^	0.25 ± 0.04 ^b^
Luteolin	1.90 ± 0.10 ^a^	1.94 ± 0.11 ^a^	0.95 ± 0.08 ^c^	1.32 ± 0.10 ^b^
Quercetin	0.63 ± 0.05 ^a^	0.60 ± 0.07 ^a^	0.29 ± 0.06 ^b^	0.16 ± 0.04 ^c^
Methoxyquercetin	0.36 ± 0.03 ^a^	0.34 ± 0.04 ^ab^	0.26 ± 0.04 ^b^	0.32 ± 0.04 ^ab^
Apigenin	0.56 ± 0.05 ^a^	0.58 ± 0.04 ^a^	0.18 ± 0.02 ^c^	0.38 ± 0.05 ^b^
Diosmetin	0.40 ± 0.05 ^a^	0.38 ± 0.04 ^a^	0.22 ± 0.03 ^c^	0.29 ± 0.04 ^b^
Trihydroxy dimethoxyflavone	0.27 ± 0.02 ^a^	0.29 ± 0.02 ^a^	0.13 ± 0.01 ^c^	0.20 ± 0.02 ^b^
Centaureidin	2.02 ± 0.12 ^a^	2.07 ± 0.09 ^a^	1.22 ± 0.05 ^c^	1.76 ± 0.08 ^b^
Methoxyacacetin	0.25 ± 0.03 ^a^	0.26 ± 0.02 ^a^	0.09 ± 0.02 ^c^	0.16 ± 0.02 ^b^
Dihydroxy trimethoxyflavone	0.44 ± 0.05 ^a^	0.46 ± 0.06 ^a^	0.17 ± 0.03 ^c^	0.31 ± 0.05 ^b^
Casticin	2.93 ± 0.10 ^a^	2.92 ± 0.11 ^a^	1.45 ± 0.09 ^c^	2.31 ± 0.10 ^b^

^a, b, c^ Different letters denote statistical differences within a line according to Fisher’s least significant difference (LSD) procedure (*p* < 0.05).

**Table 2 molecules-27-08254-t002:** Phenolic compounds (mg/g extract) of yarrow enriched-extract (EE) before and after oral, gastric and intestinal digestion steps.

Compound	Undigested-EE	Oral	Gastric	Intestinal
Neochlorogenic acid	0.15 ± 0.03 ^c^	0.15 ± 0.02 ^c^	0.22 ± 0.04 ^b^	0.86 ± 0.06 ^a^
Protocatechuic acid	0.13 ± 0.02 ^b^	0.13 ± 0.03 ^b^	0.14 ± 0.03 ^b^	0.74 ± 0.07 ^a^
Caftaric acid isomer	0.15 ± 0.02 ^a^	0.13 ± 0.02 ^a^	0.05 ± 0.01 ^b^	0.05 ± 0.02 ^b^
Caftaric acid	0.19 ± 0.06 ^b^	0.18 ± 0.05 ^b^	0.30 ± 0.08 ^a^	0.30 ± 0.06 ^a^
Caffeoylquinic acid isomer I	0.46 ± 0.07 ^a^	0.42 ± 0.06 ^a^	0.48 ± 0.07 ^a^	0.39 ± 0.06 ^a^
Chlorogenic acid	7.60 ± 0.35 ^a^	7.46 ± 0.21 ^a^	7.49 ± 0.01 ^a^	6.28 ± 0.25 ^b^
Cryptochlorogenic acid	0.12 ± 0.01 ^b^	0.14 ± 0.02 ^b^	0.15 ± 0.02 ^b^	1.11 ± 0.07 ^a^
Vicenin 2	3.20 ± 0.12 ^b^	3.22 ± 0.10 ^b^	3.37 ± 0.10 ^a^	3.49 ± 0.11 ^a^
Caffeoylquinic acid isomer II	0.22 ± 0.02 ^a^	0.24 ± 0.02 ^a^	0.27 ± 0.03 ^a^	0.28 ± 0.02 ^a^
Apigenin hexoside-pentoside I	0.76 ± 0.04 ^b^	0.77 ± 0.05 ^b^	0.96 ± 0.08 ^a^	0.80 ± 0.07 ^ab^
Caffeic acid	0.90 ± 0.06 ^a^	0.91 ± 0.06 ^a^	0.90 ± 0.04 ^a^	0.94 ± 0.05 ^a^
Schaftoside isomer	2.33 ± 0.14 ^b^	2.29 ± 0.11 ^b^	2.62 ± 0.12 ^a^	2.77 ± 0.10 ^a^
Schaftoside	3.64 ± 0.10 ^b^	3.57 ± 0.11 ^b^	4.02 ± 0.15 ^a^	3.92 ± 0.12 ^a^
Homoorientin	6.31 ± 0.21 ^a^	6.03 ± 0.16 ^a^	6.36 ± 0.18 ^a^	5.50 ± 0.13 ^b^
Apigenin hexoside-pentoside II	1.90 ± 0.10 ^a^	1.76 ± 0.09 ^a^	1.88 ± 0.08 ^a^	1.87 ± 0.09 ^a^
Luteolin dihexoside I	7.68 ± 0.19 ^a^	7.35 ± 0.12 ^b^	7.20 ± 0.10 ^b^	7.56 ± 0.11 ^ab^
6-hydroxyluteolin-7-*O*-glucoside	6.46 ± 0.20 ^a^	6.25 ± 0.16 ^a^	6.58 ± 0.21 ^a^	5.26 ± 0.18 ^b^
Apigenin dihexoside	0.44 ± 0.08	0.41 ± 0.06	0.52 ± 0.06	0.42 ± 0.04
Quercetin hexoside	4.10 ± 0.20 ^a^	4.00 ± 0.14 ^a^	3.37 ± 0.15 ^b^	0.96 ± 0.10 ^c^
Luteolin dihexoside II	0.63 ± 0.05 ^b^	0.66 ± 0.04 ^b^	0.78 ± 0.06 ^a^	0.69 ± 0.04 ^ab^
Rutin	2.86 ± 0.11 ^b^	3.19 ± 0.12 ^a^	3.30 ± 0.10 ^a^	3.00 ± 0.13 ^ab^
Apigenin hexoside	1.75 ± 0.08 ^b^	2.06 ± 0.10 ^a^	2.25 ± 0.11 ^a^	2.24 ± 0.10 ^a^
Vitexin	2.51 ± 0.10 ^a^	2.44 ± 0.09 ^a^	2.52 ± 0.10 ^a^	2.42 ± 0.08 ^a^
Apigenin hexoside- deoxyhexoside	0.85 ± 0.04 ^a^	0.89 ± 0.05 ^a^	0.72 ± 0.04 ^b^	0.56 ± 0.03 ^c^
Apigenin derivative	6.44 ± 0.21 ^c^	6.80 ± 0.22 ^c^	7.60 ± 0.21 ^b^	8.23 ± 0.30 ^a^
Luteolin-7-*O*-glucoside	24.2 ± 1.30 ^a^	23.6 ± 1.12 ^a^	19.5 ± 1.06 ^c^	21.8 ± 1.02 ^b^
Luteolin-7-*O*-glucuronide	1.57 ± 0.08 ^a^	1.45 ± 0.07 ^a^	1.06 ± 0.06 ^c^	1.17 ± 0.09 ^b^
Quercetin hexuronide	0.95 ± 0.06 ^a^	0.88 ± 0.05 ^a^	0.87 ± 0.04 ^a^	0.97 ± 0.02 ^a^
3,4-Dicaffeoylquinic acid	3.78 ± 0.18 ^b^	3.73 ± 0.12 ^b^	3.80 ± 0.10 ^b^	20.9 ± 1.22 ^a^
Isorhamnetin hexoside I	3.36 ± 0.10 ^a^	3.39 ± 0.09 ^a^	3.57 ± 0.11 ^a^	3.50 ± 0.12 ^a^
1,5-Dicaffeoylquinic acid	4.29 ± 0.27 ^ab^	4.75 ± 0.21 ^a^	4.10 ± 0.12 ^b^	3.62 ± 0.14 ^c^
3,5-Dicaffeoylquinic acid	72.4 ± 2.92 ^a^	72.5 ± 1.91 ^a^	60.4 ± 2.10 ^b^	24.2 ± 1.33 ^c^
Apigenin-7-*O*-glucoside	7.30 ± 0.33 ^a^	7.11 ± 0.21 ^a^	7.00 ± 0.18 ^a^	6.28 ± 0.15 ^b^
Luteolin-*O*-malonylglucoside	1.08 ± 0.08 ^a^	1.05 ± 0.05 ^a^	1.08 ± 0.07 ^a^	1.09 ± 0.08 ^a^
4,5-Dicaffeoylquinic acid	13.3 ± 0.87 ^b^	12.6 ± 0.63 ^b^	10.5 ± 0.51 ^c^	36.9 ± 1.21 ^a^
Isorhamnetin hexoside II	1.57 ± 0.10 ^b^	1.61 ± 0.09 ^b^	1.27 ± 0.07 ^c^	1.73 ± 0.09 ^a^
Dicaffeoylquinic acid isomer	0.26 ± 0.04 ^a^	0.28 ± 0.05 ^a^	0.23 ± 0.03 ^a^	0.23 ± 0.05 ^a^
Feruloylcaffeoylquinic acid	0.29 ± 0.05 ^a^	0.30 ± 0.06 ^a^	0.15 ± 0.03 ^b^	0.25 ± 0.04 ^a^
Tricaffeoylquinic acid	0.86 ± 0.08 ^a^	0.79 ± 0.06 ^a^	0.15 ± 0.02 ^c^	0.60 ± 0.06 ^b^
Luteolin	3.33 ± 0.15 ^a^	3.17 ± 0.11 ^a^	1.78 ± 0.09 ^c^	2.57 ± 0.10 ^b^
Quercetin	0.89 ± 0.06 ^a^	0.86 ± 0.06 ^a^	0.50 ± 0.09 ^b^	0.35 ± 0.05 ^c^
Methoxyquercetin	0.83 ± 0.08 ^a^	0.80 ± 0.07 ^a^	0.61 ± 0.06 ^b^	0.75 ± 0.07 ^a^
Apigenin	0.39 ± 0.04 ^a^	0.39 ± 0.05 ^ab^	0.12 ± 0.01 ^c^	0.31 ± 0.03 ^b^
Diosmetin	0.24 ± 0.02 ^a^	0.24 ± 0.03 ^a^	0.16 ± 0.03 ^b^	0.23 ± 0.03 ^a^
Trihydroxy dimethoxyflavone	0.35 ± 0.03 ^a^	0.35 ± 0.04 ^a^	0.18 ± 0.02 ^b^	0.31 ± 0.03 ^a^
Centaureidin	3.34 ± 0.14 ^a^	3.30 ± 0.15 ^a^	2.09 ± 0.13 ^c^	2.93 ± 0.17 ^b^
Methoxyacacetin	0.23 ± 0.03 ^a^	0.23 ± 0.02 ^a^	0.06 ± 0.01 ^c^	0.20 ± 0.02 ^a^
Dihydroxy trimethoxyflavone	0.34 ± 0.04 ^a^	0.32 ± 0.05 ^a^	0.16 ± 0.02 ^c^	0.20 ± 0.03 ^b^
Casticin	4.18 ± 0.17 ^a^	4.02 ± 0.14 ^a^	2.27 ± 0.11 ^c^	3.62 ± 0.12 ^b^

^a, b, c^ Different letters denote statistical differences within a line according to Fisher’s least significant difference (LSD) procedure (*p* < 0.05).

**Table 3 molecules-27-08254-t003:** Total phenolic content (TPC) and antioxidant activity (TEAC value) of yarrow ultrasound-assisted extract (YE) and yarrow enriched-extract (EE) after oral, gastric, and intestinal digestion.

		Undigested	Oral	Gastric	Intestinal
TPC ^1^	YE	105 ± 3 ^a^	96 ± 3 ^b^	87 ± 2 ^c^	91 ± 2 ^b^
	EE	224 ± 3 ^a^	214 ± 2 ^b^	201 ± 3 ^d^	208 ± 3 ^c^
TEAC value ^2^	YE	0.36 ± 0.01 ^a^	0.35 ± 0.01 ^a^	0.20 ± 0.01 ^b^	0.22 ± 0.04 ^b^
	EE	1.12 ± 0.06 ^a^	1.06 ± 0.03 ^a^	0.75 ± 0.06 ^b^	0.83 ± 0.04 ^b^

^1^ TPC = mg GAE/g extract. ^2^ TEAC value = mmol Trolox/g extract. ^a,b,c,d^ Different letters denote statistical differences within a same line, according to Fisher’s least significant difference (LSD) procedure (*p* < 0.05).

**Table 4 molecules-27-08254-t004:** Phenolic compounds (mg/L of digested extract) detected in the apical compartment, Caco-2 cell monolayer and basolateral compartment at 2, 4 and 6 h incubation with digested yarrow enriched-extract (EE).

	Apical Compartment			Cell Monolayer			Basolateral Compartment		
Compounds	2 h	4 h	6 h	2 h	4 h	6 h	2 h	4 h	6 h
Apigenin derivative	18.50 ± 0.04 ^a^	18.30 ± 0.07 ^b^	17.70 ± 0.10 ^c^	0.37 ± 0.0 ^a^	0.37 ± 0.01 ^a^	0.36 ± 0.01 ^a^	0.45 ± 0.01 ^a^	0.46 ± 0.01 ^a^	0.42 ± 0.01 ^b^
Luteolin-7-*O*-glucoside	41.64 ± 0.10 ^a^	40.98 ± 0.30 ^a^	39.77 ± 0.04 ^b^	0.39 ± 0.02 ^a^	0.36 ± 0.01 ^a^	0.28 ± 0.01 ^b^	0.25 ± 0.01 ^a^	0.24 ± 0.01 ^a^	0.21 ± 0.01 ^b^
3,4-Dicaffeoyl- quinic acid	14.85 ± 0.10 ^a^	12.09 ± 0.10 ^b^	9.18 ± 0.05 ^c^	0.68 ± 0.01 ^a^	0.66 ± 0.02 ^a^	n.d.	0.76 ± 0.01 ^a^	0.71 ± 0.01 ^b^	n.d.
3,5-Dicaffeoyl- quinic acid	4.95 ± 0.12 ^a^	4.26 ± 0.05 ^b^	2.99 ± 0.07 ^c^	0.75 ± 0.01	n.d.	n.d.	0.71 ± 0.01 ^a^	0.70 ± 0.01 ^a^	n.d.
Apigenin 7-*O*-glucoside	13.34 ± 0.06 ^a^	12.06 ± 0.08 ^b^	11.58 ± 0.15 ^c^	0.24 ± 0.01 ^a^	0.2 ± 0.01 ^ab^	0.20 ± 0.01 ^b^	0.14 ± 0.01	n.d.	n.d.
4,5-Dicaffeoyl- quinic acid	22.24 ± 0.14 ^a^	20.74 ± 0.11 ^b^	16.28 ± 0.11 ^c^	0.08 ± 0.01	n.d.	n.d.	0.26 ± 0.01 ^a^	0.13 ± 0.01 ^b^	0.09 ± 0.02 ^c^
Apigenin	0.16 ± 0.01 ^a^	0.09 ± 0.02 ^b^	n.d.	0.04 ± 0.01	n.d.	n.d.	0.13 ± 0.01 ^c^	0.18 ± 0.01 ^b^	0.27 ± 0.01 ^a^
Diosmetin	0.95 ± 0.02 ^a^	0.94 ± 0.03 ^a^	0.94 ± 0.01 ^a^	0.89 ± 0.01 ^a^	0.66 ± 0.01 ^b^	0.41 ± 0.01 ^c^	0.40 ± 0.01 ^c^	0.79 ± 0.02 ^b^	1.53 ± 0.11 ^a^
Centaureidin	3.27 ± 0.02 ^a^	2.18 ± 0.07 ^b^	0.75 ± 0.01 ^c^	0.58 ± 0.01 ^a^	0.41 ± 0.01 ^b^	n.d.	0.48 ± 0.01 ^c^	0.84 ± 0.07 ^b^	1.01 ± 0.02 ^a^
Methoxyacacetin	0.05 ± 0.01 ^a^	n.d.	n.d.	0.13 ± 0.01 ^a^	0.10 ± 0.0 ^b^	0.10 ± 0.01 ^b^	0.12 ± 0.01 ^c^	0.17 ± 0.01 ^b^	0.20 ± 0.02 ^a^
Casticin	5.17 ± 0.11 ^a^	4.44 ± 0.09 ^b^	3.73 ± 0.07 ^c^	1.87 ± 0.01 ^a^	1.53 ± 0.01 ^b^	1.08 ± 0.01 ^c^	1.77 ± 0.02 ^c^	2.3 ± 0.02 ^b^	3.43 ± 0.06 ^a^

^a, b, c^ Different letters denote statistical differences within a line according to Fisher’s least significant difference (LSD) procedure (*p* < 0.05). n.d.: not detected.

## Data Availability

The data presented in this study are available in this manuscript.

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
