# Peer review of "Bioavailability Assessment of Yarrow Phenolic Compounds Using an In Vitro Digestion/Caco-2 Cell Model: Anti-Inflammatory Activity of Basolateral Fraction"

_molecules, 2022, doi:10.3390/molecules27238254_

Round 1
Reviewer 1 Report
1. HPLC-PDA chromatogram should be provided at least in supplementary
2. MS/MS for compounds identification should be provided at least in supplementary
3. Caco-2 cell microscopic view should be provided to prove that has successfully established the cell culture.
4. Papp, app should be lower case. Please change it throughout the whole manuscript.
Author Response
Reviewer 1
Comments and Suggestions for Authors.
Thanks to the reviewer for his comments and suggestions that will surely help to improve the quality and understanding of the final manuscript.
- HPLC-PDA chromatogram should be provided at least in supplementary.
The HPLC-PAD base peak chromatogram (λ= 320 nm) of yarrow enriched-extract before (red line) and after (blue line) the simulated in vitro digestion process was added in the supplementary material as Figure S1. Details were included in the manuscript (Lines 98-99).
- MS/MS for compounds identification should be provided at least in supplementary.
Table S1: Phenolic compounds identified in yarrow samples by using HPLC-PAD-ESI-QTOF-MS/MS in negative ionization mode, can be found in the supplementary material as the reviewer suggested. For a better clarification of the phenolic compounds identification and quantification, some details were included in the manuscript in section 2 Results and discussions (Lines 88-92), and in section 3 Material and methods (Lines 360-372).
- Caco-2 cell microscopic view should be provided to prove that has successfully established the cell culture.
We thank you for your valuable suggestion. Nevertheless, to ensure the establishment of the cell culture we have followed the widely used and studied protocols, such as the measurement of transepithelial electrical resistance (TEER) and a complementary methodology by using a fluorescent marker, called lucifer yellow. Measurement of the TEER and the use of the lucifer yellow assay are extensively accepted methods to prove the integrity of Caco-2 cells monolayer (Uchida et al., 2009; Zeng et al., 2017; Fang et al., 2017; Villalva et al., 2018; Xu et al., 2022; Massarioli et al., 2023).
References:
Uchida, M., Fukazawa, T., Yamazaki, Y., Hashimoto, H., Miyamoto, Y. A modified fast (4 day) 96-well plate Caco-2 permeability assay. J. Pharmacol. Toxicol. Methods. 2009, 59(1), 39-43.
Zeng, Z., Shen, Z.L., Zhai, S., Xu, J.L., Liang, H., Shen, Q., Li, Q.Y., 2017. Transport of curcumin derivatives in Caco-2 cell monolayers. Eur. J. Pharm. Biopharm. 2017, 117, 123-131.
Fang, Y., Cao, W., Xia, M., Pan, S., Xu, X. Study of structure and permeability relationship of flavonoids in Caco-2 cells. Nutrients. 2017, 9(12), 1301.
Villalva, M., Jaime, L., Aguado, E., Nieto, J.A., Reglero, G., S. Santoyo. Anti-inflammatory and antioxidant activities from the basolateral fraction of Caco-2 cells exposed to a rosmarinic acid enriched extract. J. Agric. Food Chem. 2018, 66(5), 1167-1174
Xu, C., Kong, L., Tian, Y. Investigation of the Phenolic Component Bioavailability Using the In Vitro Digestion/Caco-2 Cell Model, as well as the Antioxidant Activity in Chinese Red Wine. Foods. 2022, 11(19), 3108.
Massarioli, A.P., de Oliveira Sartori, A.G., Juliano, F.F., do Amaral, J.E.P.G., Dos Santos, R.C., de Lima, L.M., de Alencar, S.M. Simulated gastrointestinal digestion/Caco-2 cell model to predict bioaccessibility and intestinal permeability of p-coumaric acid and p-coumaroyl derivatives in peanut. Food Chem. 2023, 400, p.134033.
- Papp, app should be lower case. Please change it throughout the whole manuscript.
The app lower case has been corrected throughout the whole manuscript.
Reviewer 2 Report
Marisol Villalva and team performed the Bioavailability assessment of yarrow phenolic compounds using an in vitro digestion/ Caco-2 cell model: anti-inflammatory activity of basolateral fraction. The work seems to be new and efficient and can be accepted for publication in Molecules. A careful observation of Journals format and typographical errors is suggested:
1. Some typo errors are observed, carefully remove them.
2. In the text we can’t find the description of Figure 1. So mention figure 1 in the text for clear understanding. Figure 1 also needs some improvements; label it properly and check the measurement units.
3. Graphical or schematic representation should be included for the clear and quick understanding of researchers.
4. Label the equation.
5. Unable to download the supporting material from:www.mdpi.com/xxx/s1, please check it
6. Reference style should be uniform. And add some latest references.
7. Supporting information should be the part of text in draft.
Author Response
Reviewer 2
Marisol Villalva and team performed the Bioavailability assessment of yarrow phenolic compounds using an in vitro digestion/ Caco-2 cell model: anti-inflammatory activity of basolateral fraction. The work seems to be new and efficient and can be accepted for publication in Molecules. A careful observation of Journals format and typographical errors is suggested:
Thanks to the reviewer for his comments and suggestions that will surely help to improve the quality and understanding of the final manuscript.
1.- Some typo errors are observed, carefully remove them.
The errors have been corrected through the manuscript.
2.- In the text we can’t find the description of Figure 1. So mention figure 1 in the text for clear understanding. Figure 1 also needs some improvements; label it properly and check the measurement units.
Figure 1 has been improved as reviewer suggested, also Figure 1 caption has been updated for a better understanding (Lines 325-330). In addition, in section 2 Results and discussions the text has been modified for a better clarification on Figure 1 description (Line 250-253).
3.- Graphical or schematic representation should be included for the clear and quick understanding of researchers.
As reviewer suggested, a schematic flowchart has been included in section 2 Materials and Methods, with the objective to summarize the main steps of the experimental procedure, as shown in Figure 2 (Lines 355-356).
- Label the equation.
The equation has been labelled (Line 438)
- Unable to download the supporting material from:www.mdpi.com/xxx/s1, please check it
Indeed, in the first version of the manuscript supporting material was not available. We forgot to delete this paragraph. In the revised manuscript, supporting material has been included and updated at the corresponding supplementary material section: Table S1: Phenolic compounds identified in yarrow samples by using HPLC-PAD-ESI-QTOF-MS/MS in negative ionization mode. Figure S1: HPLC-PAD base peak chromatogram (λ= 320 nm) of yarrow enriched-extract before (red line) and after (blue line) simulated in vitro digestion process. n.i.: non-identified compound (Lines 483-487).
6.- Reference style should be uniform. And add some latest references.
The reference section has been revised and corrected. Also, some of the references has been updated throughout the revised manuscript as reviewer suggested (#1, 3, 11, 17, 21, 23, 31, 32, 36, 37, 38).
- Supporting information should be the part of text in draft.
In the revised manuscript, supplementary material section has been included and updated (Lines 483-487). Also, details have been included through the manuscript for Table S1 (Line 90) and Figure S1 (Line 98).
Round 2
Reviewer 2 Report
Authors should upload manuscript file again because some figures were not inserted properly as those figure were floating in the text file so while saving it as a pdf file those images are not properly appearing in the manuscript file.
Author Response
1.- Authors should upload manuscript file again because some figures were not inserted properly as those figure were floating in the text file so while saving it as a pdf file those images are not properly appearing in the manuscript file.
We upload the manuscript in pdf file to avoid this problem.
